# Neoadjuvant Androgen Receptor Signaling Inhibitors before Radical Prostatectomy for Non-Metastatic Advanced Prostate Cancer: A Systematic Review

**DOI:** 10.3390/jpm13040641

**Published:** 2023-04-07

**Authors:** Takafumi Yanagisawa, Pawel Rajwa, Fahad Quhal, Tatsushi Kawada, Kensuke Bekku, Ekaterina Laukhtina, Markus von Deimling, Marcin Chlosta, Pierre I. Karakiewicz, Takahiro Kimura, Shahrokh F. Shariat

**Affiliations:** 1Department of Urology, Comprehensive Cancer Center, Medical University of Vienna, Wahringer Gurtel 43 18-20, 1090 Vienna, Austria; 2Department of Urology, The Jikei University School of Medicine, Tokyo 105-8461, Japan; 3Department of Urology, Medical University of Silesia, 41-800 Zabrze, Poland; 4Department of Urology, King Fahad Specialist Hospital, Dammam 32253, Saudi Arabia; 5Department of Urology, Okayama University Graduate School of Medicine, Dentistry and Pharmaceutical Sciences, Okayama 700-8530, Japan; 6Institute for Urology and Reproductive Health, Sechenov University, 119435 Moscow, Russia; 7Department of Urology, University Medical Center Hamburg-Eppendorf, 20251 Hamburg, Germany; 8Clinic of Urology and Urological Oncology, Jagiellonian University, 30-688 Krakow, Poland; 9Cancer Prognostics and Health Outcomes Unit, Division of Urology, University of Montreal Health Center, Montreal, QC H2X 0A9, Canada; 10Division of Urology, Department of Special Surgery, The University of Jordan, Amman 19328, Jordan; 11Department of Urology, University of Texas Southwestern Medical Center, Dallas, TX 75390, USA; 12Department of Urology, Second Faculty of Medicine, Charles University, 15006 Prague, Czech Republic; 13Department of Urology, Weill Cornell Medical College, New York, NY 10021, USA; 14Karl Landsteiner Institute of Urology and Andrology, 1090 Vienna, Austria

**Keywords:** prostate cancer, neoadjuvant therapy, androgen receptor signaling inhibitors, radical prostatectomy

## Abstract

(1) Background: Several phase II studies, including randomized controlled trials (RCTs), assessed the efficacy of adding androgen receptor signaling inhibitors (ARSIs) to androgen deprivation therapy (ADT) as a neoadjuvant treatment in patients treated with radical prostatectomy (RP) for prostate cancer (PCa). Summarizing the early results of these studies could help in designing phase III trials and patient counseling. (2) Methods: We queried three databases in January 2023 for studies that included PCa patients treated with neoadjuvant ARSI-based combination therapy before RP. The outcomes of interest were oncologic outcomes and pathologic responses, such as pathologic complete response (pCR) and minimal residual disease (MRD). (3) Results: Overall, twenty studies (eight RCTs) were included in this systematic review. Compared to ADT or ARSI alone, ARSI + ADT was associated with higher pCR and MRD rates; this effect was less evident when adding a second ARSI or chemotherapy. Nevertheless, ARSI + ADT resulted in relatively low pCR rates (0–13%) with a high proportion of ypT3 (48–90%) in the resected specimen. PTEN loss, ERG positive, or intraductal carcinoma seem to be associated with worse pathologic response. One study that adjusted for the effects of possible confounders reported that neoadjuvant ARSI + ADT improved time to biochemical recurrence and metastasis-free survival compared to RP alone. (4) Conclusions: Neoadjuvant ARSI + ADT combination therapy results in improved pathologic response compared to either alone or none in patients with non-metastatic advanced PCa. Ongoing phase III RCTs with long-term oncologic outcomes, as well as biomarker-guided studies, will clarify the indication, oncologic benefits, and adverse events of ARSI + ADT in patients with clinically and biologically aggressive PCa.

## 1. Introduction

The treatment landscape of prostate cancer (PCa) has starkly changed over the past decades [1]. Particularly, the development of androgen receptor signaling inhibitors (ARSIs) significantly improved survival outcomes in patients with metastatic PCa [1,2,3,4,5,6]. Treatment intensification, such as combining ARSIs and/or chemotherapy with androgen deprivation therapy (ADT), seems to evolve as a preferred treatment strategy for a variety of PCa states [7,8].

Neoadjuvant therapy, defined as induction therapy before local definitive treatment, is becoming widely used for various cancers, including urologic cancers [9,10]. The aims of this strategy are to reduce the primary tumor burden (thereby facilitating local definitive therapy) and eliminate possible micrometastasis that leads to disease recurrence and progression. Older studies have, however, failed to show a survival benefit to neoadjuvant ADT monotherapy in patients with clinically localized PCa before radical prostatectomy (RP) [11]. Now, there is hope that a combination of ARSIs with ADT may result in higher efficacy compared to ADT alone; however, there is still no convincing evidence for this hypothesis. Therefore, we conducted this systematic review in order to collect all the available data and assess the cumulative effect of neoadjuvant ARSI-based combination therapy on pathologic response in the RP specimen and oncologic outcomes in patients with non-metastatic advanced PCa. The comparative safety/adverse events of these strategies were also evaluated. In addition, we also report on the association of molecular/gene biomarkers and pathologic response to neoadjuvant ARSI-based combination therapy.

## 2. Materials and Methods

The protocol has been registered in the International Prospective Register of Systematic Reviews database (PROSPERO: CRD42022368246).

### 2.1. Search Strategy

The guidelines of the Preferred Reporting Items for Meta-Analyses of Observational Studies in Epidemiology Statement (PRISMA) were followed when conducting this systematic review (Appendix A) [12]. A literature search in PubMed^®^, Web of Science™, and Scopus^®^ databases was carried out in January 2023 to identify studies investigating the pathologic, oncologic, or safety outcomes of neoadjuvant ARSI-based combination therapy prior to RP. The detailed search strategy was as follows: (prostate cancer) AND (neoadjuvant) AND (prostatectomy) AND (abiraterone) OR (apalutamide) OR (enzalutamide) OR (darolutamide). In order to include unpublished randomized controlled trials (RCTs) and trial updates, we also reviewed abstracts presented at recent major conferences between 2017 and 2022, including those at the American Society of Clinical Oncology (ASCO) and the European Society for Medical Oncology. The primary outcomes of interest were the pathologic responses, such as pathological complete response (pCR) and minimal residual disease (MRD) in the resected specimen. Intratumoral hormonal alterations, treatment-emergent adverse events (TEAEs), perioperative complications, and the association between biomarkers and pathologic responses were the other measurement outcomes. Two investigators carried out the initial screening based on the titles and abstracts to find eligible studies. Potentially relevant studies were subjected to a full-text review. Disagreements were resolved by consensus with the co-authors.

### 2.2. Inclusion and Exclusion Criteria

Studies were selected if they investigated non-metastatic advanced PCa patients (Patients), who underwent neoadjuvant ARSI-based systemic combination therapy (Interventions) compared to those treated with ADT alone, other combinations, or no systemic therapy (Comparisons) to assess the differential pathologic and/or perioperative outcomes (Outcome) in RCTs, nonrandomized, observational, population-based, or cohort studies (Study design). Studies lacking original patient data, reviews, letters, editorial comments, replies from authors, case reports, and non-English-language papers were excluded. All publications included had their references checked for relevant additional research. 

### 2.3. Data Extraction

Two authors independently extracted the following data: the first author’s name, publication year, national clinical trial (NCT) number, inclusion criteria, number of patients, treatment regimen and duration, follow-up periods, age, pretreatment prostate-specific antigen (PSA), biopsy Gleason score (GS) or International Society of Urological Pathology (ISUP), Gleason grade (GG), clinical stage, D’Amico or National Comprehensive Cancer Network (NCCN) risk classification, PSA kinetics before RP, the pCR and MDR achievement rates, total tumor volume, residual cancer burden (RCB), the proportion of non-organ confined disease (ypT ≥ 3), pathological node-positively (pN+), positive surgical margins (PSMs), the rates of TEAEs (any and severe [CTCAE ≥ grade3]), perioperative complications, PSA recurrence rates, and the association of endpoints with analyzed biomarkers. All discrepancies were resolved by consensus with the co-authors. 

### 2.4. Risk of Bias Assessment

According to the Cochrane Handbook for Systematic Reviews of Interventions and the Risk of Bias in Non-randomized Studies of Interventions (ROBINS-I) tool and the risk-of-bias (RoB version2), the study’s quality and the risk of bias were evaluated [12]. The degree of each bias domain and the overall risk of bias were rated as ‘Low’, ‘Moderate’, ‘Serious’, or ‘Critical’. A literature review and a consensus were used to figure out if there were any possible confounders. Two authors independently evaluated the ROBINS-I and risk of bias assessments of each study (Appendix A).

## 3. Results

### 3.1. Study Selection and Characteristics

Our initial search identified 166 records. After removing duplicates, 137 records remained for screening titles and abstracts (Figure 1). After the screening, a full-text review of 30 articles was performed. According to our inclusion criteria, we finally identified 20 studies eligible for systematic review [13,14,15,16,17,18,19,20,21,22,23,24,25,26,27,28,29,30,31,32]. Of the twenty studies, we identified eight phase II RCTs comparing the efficacy and/or safety of ARSI-based combination therapy versus other combinations or ADT/ARSI alone (Table 1) [13,14,15,16,17,18,19,20]. We consolidated the current evidence with a focus on phase II RCTs. However, despite RCTs, inclusion criteria, treatment regimen, duration, and the definition of MRD differed across RCTs; therefore, we did not perform a pooled analysis or meta-analysis.

### 3.2. Endocrinological Outcomes

In 2014, Taplin et al. first conducted phase II RCT, assessing the endocrinologic impact of neoadjuvant abiraterone (ABI) + ADT in patients with high-risk clinically localized PCa [13]. Fifty-eight patients were randomly assigned to ABI + ADT or ADT alone for 12 weeks followed by a prostate biopsy to analyze the intraprostatic endocrinologic changes [13]. The authors showed that ABI + ADT significantly reduced intraprostatic androgen levels, such as dehydroepiandrosterone (DHEA) (*p* < 0.001), ∆^4^-androstene-3,17-dione (*p* < 0.001), dihydrotestosterone (DHT) (*p* < 0.001), and testosterone (*p* = 0.02), compared to ADT alone [13].

In 2017, Montgomery et al. conducted phase II RCT and assessed the differential pathologic and hormonal response to neoadjuvant enzalutamide (ENZ) + dutasteride (DUT) + ADT versus ENZ alone [14]. Tissue hormonal results after 6 months of neoadjuvant treatment revealed that DHEA levels were not different between the two treatment groups [14]. On the contrary, tissue DHT and testosterone were significantly higher in the ENZ arm than in the ENZ + DUT + ADT arm, reflecting the lack of a negative feedback loop to the hypothalamus [14]. In addition, the authors demonstrated that tissue testosterone and DHT levels correlated with pathologic responses, such as RCB. 

In summary, these exploratory studies demonstrated that ARSI + ADT significantly reduced intraprostatic androgen levels compared to ADT or ARSI alone.

### 3.3. Pathologic Responses

Several studies assessing the efficacy of neoadjuvant ARSIs used a pathological endpoint as a surrogate for long-term oncological outcomes (Table 2 and Appendix A) [13,14,15,17,18,19,21,22]. However, no consensus yet exists regarding the ideal definition of a pathological response following neoadjuvant hormonal therapy. As shown in Table 2, all eligible studies reported pCR and MRD rates. The definition of MRD differed across studies. Four studies defined the MRD as residual cancer <5 mm as the longest length in the crossing section dimension [13,15,17,18], and one study used the <3 mm cut-off [14], and two studies defined it as RCB < 0.25 cm^3^ [18,21]. Following the definition of combined pathologic response (pCR + MDR) reported by McKay et al. in 2021 [17], we calculated the combined pathologic response of each study. The summary of pathologic outcomes regarding the rates of pCR and the achievement of MDR is shown in Table 2.

#### 3.3.1. ARSI Monotherapy

Two studies assessed the pathological response to neoadjuvant ARSI monotherapy with disappointing results [14,22]. Montgomery et al. reported that no patients receiving neoadjuvant ENZ monotherapy achieved pCR or MRD < 3 mm [14]. The NEAR trial assessed the efficacy of neoadjuvant apalutamide (APA) monotherapy in a phase II study comprising 30 patients; no patient achieved pCR [22].

#### 3.3.2. Single ARSI Plus ADT

Based on the rationale that neoadjuvant ARSI + ADT significantly reduced intraprostatic androgens compared to ADT alone [13], pathologic responses were analyzed. Taplin et al. showed a better combined pathologic response rate (pCR + MDR) in patients treated with 6 months of ABI + ADT (23%) compared to those treated with 3 months of ADT alone followed by 3 months of ABI + ADT (3.6%) [13]. However, the authors were disappointed by the low pCR rates (10%) and the high ypT3 rates (48%) despite six months of ARSI +ADT treatment [13]. Montgomery et al. reported a favorable pathologic response to six months of neoadjuvant ENZ + DUT + ADT combination over ENZ monotherapy [14]. Still, the combined pathologic response was only 17%, with low pCR rates (4.3%) and high ypT3 rates (61%) [14]. These two studies included both intermediate- and high-risk clinically localized PCa patients with 20–24% of ≥cT3 based on magnetic resonance imaging (MRI); therefore, high rates of ypT3 (48–61%) after long-term neoadjuvant ARSI + ADT seems discouraging.

Most recently, the results from the ARNEO trial led by Devos et al. were published [18]. This is a phase II RCT assessing the efficacy of a 3-month neoadjuvant degarelix with or without APA prior to RP in 89 patients with high-risk clinically non-metastatic PCa [18]. The authors demonstrated better pathologic response with regards to MDR in patients treated with APA + ADT (38%) compared to those treated with ADT alone (9%); nevertheless, there were no patients who had pCR in the APA + ADT arm, and approximately 50% of men had ypT3 PCa [18].

Taken together, these results from phase II RCTs support the efficacy of neoadjuvant ARSI + ADT combinations in high-risk clinically localized PCa patients in terms of pathologic response, providing a hypothesis-generating basis for phase III trials evaluating time-dependent survival outcomes. However, the low pCR rates and the high proportion of ypT3 patients suggest the need for more effective treatment regimens, as well as a need for accurate biomarkers, that can help to identify the candidates who are most likely to benefit from neoadjuvant ARSI-based combination therapy. 

#### 3.3.3. Double ARSIs Plus ADT

Three phase II RCTs and one single-arm study have assessed the efficacy of double ARSIs + ADT as a neoadjuvant therapy for advanced clinically non-metastatic PCa [15,17,19,21]. The rationale for this intensified regimen is to investigate whether blocking all sources of androgen production (i.e., testes, adrenal gland, and intratumoral) and maximally blocking the androgen receptor could improve the pathologic response compared to incomplete androgen blockade. 

McKay et al. conducted two phase II RCTs assessing the pathologic response to double ARSIs + ADT in 2019 and 2021 [15,17]. The first RCT published in 2019 compared the efficacy of a 6-month neoadjuvant ABI + ENZ + ADT (n = 50) with ENZ + ADT (n = 25) [15]. The combined pathologic response (pCR + MRD) rates were 30% in the ABI + ENZ + ADT arm and 16% in the ENZ + ADT arm (*p* = 0.3) [15]. Another RCT published in 2021 compared the efficacy of a 6-month neoadjuvant APA + ABI + ADT (n = 59) with ABI + ADT (n = 59) [17]. The combined pathologic response rates were similar in both groups (22% for APA + ABI + ADT vs. 20% for ABI + ADT, *p* = 0.4) [17].

Bastos et al. presented the results from a phase II RCT of the ASCO-GU annual meeting 2022, which assessed the pathologic response to a 3-month neoadjuvant APA + ABI + ADT (n = 31) compared to ABI + ADT (n = 31) [19]. This study comprised only patients with high-risk clinically non-metastatic PCa. No statistically significant differences were seen between the two groups regarding combined pathologic responses with disappointing low rates (both 6.4%) [19]. 

In summary, current phase II RCTs have failed to demonstrate the potential benefit of maximal androgen blockade with double ARSIs + ADT before RP in terms of pathologic response compared to single ARSI + ADT. This finding implies that other signaling pathways in addition to the androgen receptor (AR) axis are likely to contribute to treatment resistance and disease progression even in the non-metastatic setting.

#### 3.3.4. Chemotherapy Plus ARSI Plus ADT

A phase II RCT, the ACDC-RP trial assessed the impact of adding cabazitaxel to ARSI + ADT on pathologic outcomes [20]. This study revealed no differences in pCR (5% for cabazitaxel + ABI +ADT and 9% for ABI + ADT) and MRD (defined as <5% of prostate volume involved by a tumor) (39% for cabazitaxel + ABI + ADT and 34% for ABI + ADT) rates [20].

### 3.4. The association of Possible Biomarkers with Pathologic Outcomes 

Several phase II RCTs examined the association of biomarkers with pathologic response. Efstathiou et al. explored biomarkers associated with the treatment regimen and residual tumors in patients treated with ABI + ADT (n = 44) or ADT alone (n = 21) [16]. Glucocorticoid receptor (GR) overexpression was more frequently seen in the ABI + ADT arm than in the ADT arm alone (*p* = 0.008). In addition, GR overexpression (defined as >10% expression in tumor cells) was associated with a higher tumor epithelium volume only within the ABI + ADT arm (*p* = 0.018) and correlated with higher intraprostatic cortisol levels [16]. 

McKay et al. performed immunohistochemistry (IHC) in 60 specimens in patients treated with ENZ + ADT with or without ABI [15]. The authors showed that residual tumors had comparable levels of ETS-related gene (ERG), phosphatase and tensin homolog (*PTEN*), AR, and GR expression [15]. Of note, tumor ERG expression and *PTEN* loss were both significantly associated with more extensive residual tumors in the RP specimen [15]. In addition, the authors reported the pooled results of the previous three phase II RCTs, including Taplin et al. in 2014 (NCT00924469), Montgomery et al. in 2017 (NCT01547299), and McKay et al. in 2019 (NCT02268175) [26]. This pooled analysis verified that *PTEN* loss (*p* = 0.012), ERG positivity (*p* = 0.022), and intraductal carcinoma (IDC) (*p* = 0.001) were associated with a decreased likelihood of pathologic response [26]. The authors confirmed this finding in another phase II RCT comparing APA + ABI + ADT with ABI + ADT [17]. 

Predicting treatment response prior to neoadjuvant therapy was assessed in the ARNEO trial by Devos et al., who demonstrated that *PTEN* loss in the initial prostate biopsy was associated with significantly less MRD (*p* = 0.002) and a higher residual cancer burden (RCB, *p* < 0.001) in the RP specimen compared to those without *PTEN* loss [18]. Another pilot study by Wilkinson et al., including 37 patients treated by ENZ + ADT, demonstrated that *PTEN* loss, *TP53* alterations, ERG expression on IHC, and the presence of IDC in the initial prostate biopsy are associated with poor pathologic response defined as 0.05 cm^3^ for RCB [27]. Tewari et al. performed whole-exome and transcriptome sequencing using an initial multi-regional biopsy specimen to examine the possible molecular biomarkers to predict exceptional responders (defined as other than non-responders, such as ypT3 or pN+) [28]. The authors showed that clonal *TP53* mutation and *PTEN* copy-number loss are observed exclusively in non-responders [28].

Expression of the androgen receptor splice variant (AR-V7) has been suggested to partake in resistance mechanisms in the metastatic castration-resistant PCa setting [33]. AR-V7 was reported to be upregulated in patients with clinically localized high-risk PCa [34]. Efstathiou et al. showed that the presence of nuclear AR-V7 correlated with residual cancer burden in the resected specimen in patients treated with a 3-month neoadjuvant ADT ± ABI [16]. Conversely, another pilot study reported that while AR-V7 expression was detected in all 16 included patients with clinically localized high-risk PCa, prior to receiving neoadjuvant ABI + bicalutamide + ADT, its level of expression was not correlated with pathologic response [23]. These contradicting findings suggest that the potential role of AR-V7 as a predictive biomarker for response to neoadjuvant ARSI-based therapy remains to be studied.

Taken together, *PTEN* loss and *TP53* alteration, as well as positive ERG and IDC, seem promising biomarkers for predicting pathologic response to neoadjuvant ARSI-based therapy, possibly helping advance the concept of biomarker-driven trials ushering in the age of precision medicine.

### 3.5. Oncologic Outcomes after Neoadjuvant ARSI-Based Therapy Followed by RP

To date, there are no phase III RCTs reporting clinically significant endpoints, such as metastasis-free survival (MFS), cancer-specific survival (CSS), or overall survival (OS) in patients treated with neoadjuvant ARSI-based combination therapy followed by surgery. In addition, studies reporting biochemical recurrence (BCR) rates after neoadjuvant ARSI-based therapy followed by RP are scarce. Efstathiou et al. reported in a phase II RCT the differential rates of PSA recurrence [16]. The authors reported that 44% of patients in the ABI + ADT group versus 59% in the ADT alone group developed BCR over a 4-year follow-up period (*p* = 0.28); while a 15% difference seems clinically significant, the study was underpowered [16]. In addition, the authors showed that lower tumor epithelium volume correlated with improved BCR-free survival at a follow-up of 4 years (*p* = 0.001) [16]

Of note, McKay et al. reported the pooled analyses of distant oncologic outcomes in patients treated with neoadjuvant ARSI-based combination therapy followed by RP from three phase II RCTs, including Taplin et al. in 2014 (NCT00924469), Montgomery et al. in 2017 (NCT01547299), and McKay et al. in 2019 (NCT02268175) [26]. Overall, 117 patients receiving neoadjuvant ARSI-based combination therapy were eligible for analysis, with 49 (42%) and 15 (13%) patients developing BCR and metastasis, respectively [26]. The 3-year BCR-free and 5-year MFS rates were 59.1% (95% confidence interval [CI]: 49.0–67.9) and 87.8% (95% CI: 76.4–93.9%), respectively [26]. Notably, of the twenty-five patients with exceptional pathological response, only two (8.0%) developed BCR, but no patient developed metastasis and cancer death during a median follow-up of 3.6 years [26]. The authors verified that patients with *PTEN* loss and IDC in the RP specimen had a shorter time to BCR compared to those without these biomarker alterations [26]. A recently published comparative study led by Ravi et al. assessed the differential oncologic outcomes between ARSI-based combination neoadjuvant therapy followed by RP versus RP alone, using the cohort from the aforementioned three phase II RCTs as the intervention arm and a control cohort of patients who met eligibility criteria from their institution [32]. After matching for the effect of possible confounders using an inverse probability of treatment weighting (IPTW) methods, time to BCR (HR: 0.25, 95% CI: 0.18–0.37) and MFS (HR: 0.26, 95% CI: 0.15–0.46) were significantly longer in patients treated with ARSI-based combination neoadjuvant therapy compared to those who underwent RP only [32].

Recently, results from the ACDC-RP trial, which assessed whether adding cabazitaxel improves pathologic and/or oncologic outcomes, revealed no difference in pathologic response and BCR-free survival rates between cabazitaxel + ABI + ADT and ABI + ADT [20]. Nevertheless, this study confirmed the previous findings suggesting that patients who achieved exceptional pathologic response experience longer BCR-free survival compared to those who did not [20].

Despite the lack of phase III RCTs, a pooled analysis of phase II RCTs showed consistently superior oncologic outcomes of ARSI-based combination neoadjuvant therapy compared to patients who underwent RP only. Patients who obtained a deep pathologic response to neoadjuvant therapy had a better prognosis, suggesting that neoadjuvant therapy with meticulous pathologic and molecular evaluation of the RP specimen can help us identify those patients with biologically and clinically aggressive disease that requires additional intensified treatment.

### 3.6. Radiographic Assessment of Treatment Efficacy in Patients Treated with Neoadjuvant Hormonal Therapy 

The radiographic evaluation of treatment efficacy during neoadjuvant therapy is necessary to assess the success/failure of this treatment strategy. In 2019, Gold et al. conducted a phase II study to assess the diagnostic performance of multiparametric MRI (mpMRI) to evaluate/estimate disease severity and extent in 20 patients treated with neoadjuvant ENZ + ADT [25]. The authors showed a satisfactory positive predictive value of extraprostatic extension (71%), seminal vesicle invasion (80%), and organ-confined disease (80%) [25]. However, a phase II RCT by McKay et al. in 2021 reported a low concordance and correlation between mpMRI findings after neoadjuvant ARSI-based combination therapy and pathological residual tumor volume and pCR [17]. Seventy-one patients who received ABI + ADT with or without APA had a central review of their mpMRI images; while thirteen patients (18%) were staged as a complete response on mpMRI, and only one had pCR [17]. 

In 2021, Chen et al. conducted a pilot study assessing the performance of a 68Ga-prostate-specific membrane antigen (PSMA)-11 positron emission tomography (PET)/CT in the evaluation of treatment with neoadjuvant ABI + ADT for high-risk clinically localized PCa [24]. The authors showed that PET/CT changes had higher specificity in the assessment of pathologic response than PSA changes (89.7% vs. 62.1%, *p* = 0.043) [27]. In addition, using a multivariable analysis, only the high post-treatment maximum standardized uptake (SUVmax) value was an independent predictor of worse pathologic response [27].

In the ARNEO trial, 18F-PSMA-1007 PET/MRI was performed before and after neoadjuvant therapy [18]. This study demonstrated that PSMA-PET estimated tumor volumes and SUVmax values, which were significantly lower in patients with MRD (RCB < 0.25 cm^3^) in the resected specimen compared to those without MRD [18]. In line with this, Bastos et al. showed that patients with complete PSMA-PET response (50%) had a higher rate of RCB < 0.25 cm^3^ compared to those without complete PSMA-PET response (7.5%, *p* = 0.001) [19]. Of note, during a median follow-up of 2.6 years, all patients with both complete PSMA-PET response and RCB < 0.25 cm^3^ remained BCR-free [19].

In summary, novel imaging modalities, such as PSMA-PET/CT or MRI, appear to achieve good diagnostic performance for predicting pathologic response, suggesting that future studies need to incorporate the pre- and post-treatment evaluation using PSMA-PET/CT or MRI. In addition, Bright et al. recently demonstrated the diagnostic utility of IHC with antibodies against PSMA for detecting residual tumors in patients treated with 6 months of neoadjuvant therapy with ENZ + ADT, supporting the importance of PSMA both pathologically and radiographically [30].

### 3.7. Safety

ARSIs have agent-specific adverse events (AEs) with a general benefit–harm balance needing to be considered for clinical application, especially when considered in the non-metastatic setting. There are two kinds of AEs needing consideration: one is treatment-emergent AEs (TEAEs) (i.e., directly related to ARSI therapy) and the other is perioperative complications due to potentially increased technical difficulty of surgy after ARSI + ADT (i.e., severe adhesion). The rates of TEAEs and perioperative complications are summarized in Table 3.

#### 3.7.1. Treatment-Emergent Adverse Events

Similar to the metastatic PCa setting, Efstathiou et al. reported that ABI + ADT (39%) increased the risk of severe TEAEs compared to ADT alone (24%), with 11% treatment discontinuation rates in the ABI + ADT group [16]. The ARNEO trial showed that 8.9% of patients suffered severe rash in the APA + ADT group [18].

Regarding double ARSIs + ADT treatment, adding ABI to ENZ + ADT significantly increased the risk of severe hypertension (10% vs. 0%) and increased transaminase (10% vs. 0%) compared to ENZ + ADT [15]. Two phase II RCTs reported that adding APA to ABI + ADT also increased the risk of severe TEAEs compared to ABI + ADT [17,19]. Despite a limited number of patients included in each RCT, double ARSIs + ADT seems to be associated with an increased risk of severe TEAEs compared to single ARSI + ADT.

#### 3.7.2. Perioperative Complications

As shown in Table 3, phase II RCTs reported comparative perioperative complication rates between treatment and control arms. However, as patients in the control arm also received neoadjuvant hormonal therapy, the potential impact of ARSI-based combination therapy on perioperative outcomes is still unclear. Recently, Ilario et al. conducted a comparative study assessing the differential perioperative complication rates between patients treated with ARSI-based neoadjuvant combination therapy and those without [31]. The patients (n = 61) in the neoadjuvant ARSI group were from a phase II RCT (NCT02789878), and the patients (n = 63) who did not receive neoadjuvant therapy were not included in the RCT but received therapy during the same period [31]. The authors showed no significant differences in perioperative complication rates between the two groups [31]. Another three-arm phase II RCT led by Sterling et al. assessed the feasibility of nerve-sparing during RP after intensified ARSI-based neoadjuvant therapy in patients with high-risk localized PCa (n = 24) [29]. The authors reported on the technical feasibility of performing a nerve-sparing approach specifically owing to the reduced tumor volume after ARSI-based neoadjuvant therapy; this was not associated with reduced potency [29].

Based on current literature, the risk of perioperative complications seems not to increase after neoadjuvant ARSI-based combination therapy. However, the small cohort size makes a reliable conclusion challenging.

## 4. Discussion and Future Perspective

In this systematic review, we summarized the current evidence regarding ARSI-based neoadjuvant therapy prior to RP for non-metastatic advanced PCa. We had to rely on multiple phase II RCTs and pilot prospective studies, challenging reliable and robust conclusions. Although neoadjuvant ARSI-based combination therapy reliably results in a pathologic response with possible biomarkers of a response having been identified, further investigation with a long-term follow-up is needed to elucidate the clinically relevant endpoints. In addition, other possible combinations, such as chemohormonal therapy and/or other definitive local therapy (i.e., radiation therapy [RT]), are needed to discuss a comprehensive concept of intensified treatment for non-metastatic advanced PCa. 

The utility of treatment intensification, such as perioperative systemic therapy, in addition to definitive local therapy for non-metastatic locally advanced PCa, has been demonstrated previously [7]. Especially, as a part of intensified treatment, the utility of perioperative chemohormonal therapy, that is, docetaxel plus ADT, has been reported [35,36,37,38]. Notably, a phase III RCT comprising 738 localized high-risk PCa patients conducted by Eastham et al. showed that neoadjuvant docetaxel + ADT improved MFS (HR: 0.70, 95% CI: 0.51–0.95) and OS (HR: 0.61, 95% CI: 0.40–0.94) compared to RP alone [36]. In addition, a recent meta-analysis supported that perioperative chemohormonal therapy followed by definitive local therapy (RT and RP) improves CSS (pooled HR: 0.68, 95% CI: 0.49–0.95) and MFS (pooled HR: 0.82, 95% CI: 0.71–0.95). In a sensitivity analysis excluding the study of Eastham et al., there was some evidence of improved survival in patients treated with docetaxel + ADT, but it did not reach statistical significance [7]. 

When focusing on the survival impact of perioperative ARSI-based combination therapy, the STAMPEDE trial, which compared perioperative ARSI (ABI ± ENZ) + ADT combination versus ADT alone, in addition to radiation therapy (RT) for high-risk non-metastatic PCa, showed that ARSI-based combinations significantly improve OS (HR: 0.60, 95% CI: 0.48–0.73) [39]. An aforementioned meta-analysis demonstrated that ARSI-based combination therapy outperformed docetaxel + ADT in terms of all survival endpoints in patients who underwent RT using a network meta-analysis [7]. Together with the results from previous studies, neoadjuvant ARSI-based combination therapy followed by RP can be a promising treatment strategy for non-metastatic advanced PCa. Although Ravi et al. recently reported that ARSI-based neoadjuvant therapy significantly improved the time to BCR and MFS compared to RP alone, further investigation with a well-designed phase III RCT is, indeed, urgently needed.

The PROTEUS trial, the first phase III RCT, which aimed to assess the efficacy (primary endpoints were pCR and MFS) of a 6-month neoadjuvant ADT + APA before RP followed by a 6-month adjuvant ADT + APA versus ADT alone with 2000 patients, is ongoing [40]. However, although neoadjuvant ADT has never shown a survival benefit compared to RP alone, most ongoing RCTs set the control arm as neoadjuvant ADT only. To date, the standard of care for high-risk non-metastatic PCa is RP alone when surgical treatment is applied [1]. The results from this RCT will provide novel insight into the efficacy of ARSI-based neoadjuvant therapy for non-metastatic advanced PCa, while interpretation may be controversial.

Novel maximum androgen blockade using conventional ADT with an androgen-synthesis inhibitor (i.e., ABI) and an AR antagonist (i.e., ENZ or APA) can hypothetically obtain preferable oncologic outcomes compared to an incomplete androgen blockade. Therefore, this intensified treatment regimen has been tested in several PCa settings. The STAMPEDE trial compared the distant oncologic outcomes in patients with high-risk localized PCa treated with ABI + ENZ + ADT versus ABI + ADT [39]. However, this study showed no differences in MFS between ABI + ENZ + ADT and ABI + ADT (HR: 1.02, 95% CI: 0.70–1.50) [39]. In addition, in the first-line metastatic castration-resistant PCa setting, the ACIS trial failed to show an OS benefit with APA + ABI + ADT compared to ABI + ADT (HR: 0.95, 95% CI: 0.81–1.11) [41]. Therefore, a novel maximum androgen blockade has still not been applied in clinical practice. Based on discouraging pathologic response and the increased risk of TEAEs, this intensified treatment regimen with double ARSIs + ADT seems suboptimal for the neoadjuvant setting.

Finally, the optimal treatment duration of neoadjuvant therapy needs to be considered. Included phase II RCTs set the treatment duration as three or six months. For a comparison of the same treatment regimen (APA + ABI + ADT vs. ABI + ADT), Mackay et al. studied a 6 months of neoadjuvant therapy, while Bastos et al. studied the 3-month strategy [17,19]. Despite some differences in patient demographics, the authors reported pCR rates of 13% and 10% for 6 months of treatment in the APA + ABI + ADT and ABI + ADT arms compared to 3.2% and 0% for 3 months of treatment [17,19]. In patients treated with neoadjuvant ADT alone before RP, a recent meta-analysis showed that long-term neoadjuvant ADT was associated with more favorable pathologic outcomes, while the impact of treatment duration on survival outcomes remains unproven due to limited evidence [42]. Therefore, further investigation is needed to clarify the optimal duration of neoadjuvant ARSI-based therapy in terms of survival benefit. 

Despite several controversies and issues on clinical application, current evidence suggests that neoadjuvant ARSI-based therapy achieves measurable and possibly variable pathologic response and may contribute to improving distant oncologic outcomes in patients with non-metastatic advanced PCa. In addition, current studies provided molecular analyses to help predict pathologic response in the future and uncover resistance mechanisms. 

## 5. Conclusions

Current evidence shows that neoadjuvant ARSI + ADT combinations offer favorable pathologic response compared to ADT or ARSI alone in patients with non-metastatic advanced PCa. However, triple androgen blockades, such as double ARSIs + ADT, did not improve the pathologic response compared to single ARSI + ADT. Despite the ARSI-based neoadjuvant therapy, low pCR rates and a high proportion of ypT3 in the resected specimen have been reported. Promising biomarkers for predicting the outcomes of ARSI-based neoadjuvant therapy, such as *PTEN* loss, ERG-positive, and/or the presence of IDC, could help guide future clinical trials and facilitate precision medicine strategies in this disease state.

## Figures and Tables

**Figure 1 jpm-13-00641-f001:**
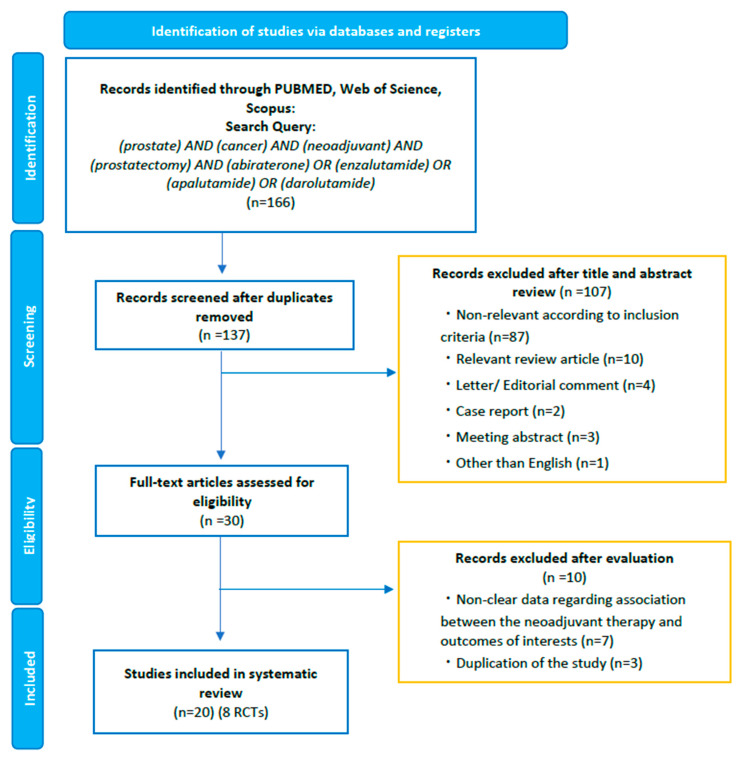
The Preferred Reporting Items for Systematic Reviews and Meta-Analyses (PRISMA) flow chart, detailing the article selection process.

**Table 1 jpm-13-00641-t001:** Study and patient demographics of the 20 included studies.

First Author	NCT Number	Year	Study Design	Outcome Measurement	Inclusion Criteria	Treatment Regimen			Treatment Duration	No. of Patients		
						I		C		Total	I	C
**RCTs assessing pathologic outcomes**												
Taplin [13]	NCT00924469	2014	Phase II	Pathologic response Safety	≥3 positive cores, any of: PSA ≥ 10 ng/mL, PSAV ≥ 2 ng/mL/yr, GS ≥ 7	ABI + ADT (24 w)		ADT (12 w) ABI + ADT (12 w)	6 M	58	30	28
Montgomery [14]	NCT01547299	2017	Phase II	Pathologic response	T1c–T3, ≥3 positive cores, GS ≥ 7, PSA > 10 ng/mL; N0M0 (BS, CT/MRI)	ENZ + DUT + ADT		ENZ	6 M	48	23	25
McKay [15]	NCT02268175	2019	Phase II	Pathologic response Safety	GS ≥ 4 + 3, ≥3 positive cores or >1 cm tumor on MRI, PSA ≥ 20 ng/mL, or T3 on MRI; N < 20 mm, M0	ABI + ENZ + ADT		ENZ + ADT	6 M	75	50	25
Efstathiou [16]	NCT01088529	2019	Phase II	Pathologic response Safety BCR	≥T1c with GS 8–10 or ≥T2b with GS 7 N0M0 (BS, CT)	ABI + ADT		ADT	3 M	65	44	21
McKay [17]	NCT02903368	2021	Phase II	Pathologic response Safety	GS ≥ 4 + 3, GS < 3 + 4 with PSA > 20 ng/mL, or T3 (MRI), ≥3 positive cores, tumor >1 cm (MRI), or T3 (MRI); N < 20 mm, M0	APA + ABI + ADT		ABI + ADT	6 M	118	59	59
Devos [18]	ARNEO NCT03080116	2022	Phase II	Pathologic response Safety	Unfavorable intermediate risk GS 7, PSA 10–20 ng/mL, and/or cT2b (MRI) or high-risk GS 8–10, PSA > 20, cT2c (MRI), and/or cN1	APA + ADT		ADT	3 M	89	45	40
Bastos [19]	NCT02789878	2022 (ASCO-GU)	Phase II	Pathologic response Safety	High-risk GS 8–10 and/or PSA > 20 and/or cT3 (MRI) and/or cN1	APA + ABI + ADT		ABI + ADT	3 M	62	31	31
Fleshner [20]	NCT02543255	2022 (ASCO-GU)	Phase II	Pathologic response Safety	High-risk (D’Amico) GS 8–10 and/or PSA > 20 or T2c-3 based on DRE +/- imaging	CBZ + ABI + ADT		ABI + ADT	3 M	70	38	32
**Single-arm studies**												
Graham [21]	ND	2021	Phase II	Pathologic response	NCCN high- to very high-risk, N0M0	Indomethacin + APA + ABI + ADT			3 M	20		
Lee [22]	NEAR	2022	Phase II	Pathologic response	D’Amico intermediate- (cT2b or PSA10–20 ng/mL or GS of 7) or high-risk (cT2c-4 or PSA > 20 ng/mL or GS: 8) N0M0 (BS, MRI, CT)	APA			3 M	30		
Corcoran [23]	ND	2015	Phase II	Pathologic response Expression of ARv7	High-risk	ABI + bicalutamide + ADT			6 M	17		
Chen [24]	NCT04356430	2021	Phase II	Predictive value of PSMA PET/CT	High-risk N0M0 ≥cT3(MRI or PSMA PET/CT) or GS 8–10 or PSA ≥ 20	ABI + ADT			6 M	45		
Gold [25]	NCT02430480	2019	Phase II	Association between mpMRI findings and pathology	Intermediate-risk: cT2b-c or GS 7 or PSA 10–20 or high-risk: ≥cT3 or GS 8–10 or PSA ≥ 20	ENZ + ADT			6 M	20		
Mckay [26]	NCT00924469 NCT01547299 NCT02268175	2021	Pooled analysis of Phase II RCTs	Time to BCR MFS, OS	Following each RCT’s eligibility criteria	ARSI + ADT			6 M	117		
Wilkinson [27]	NCT02430480	2021	Phase II	Molecular and histologic features and MRI imaging	Intermediate-risk: cT2b-c or GS 7 or PSA 10–20 or high-risk: ≥cT3 or GS 8–10 or PSA ≥ 20	ENZ + ADT			6 M	37		
Tewari [28]	NCT02268175 NCT02903368	2021	Experimental study	Molecular features on the pretreatment biopsy specimen	Intermediate-risk: cT2b-c or GS 7 or PSA 10–20 or high-risk: ≥cT3 or GS 8–10 or PSA ≥ 20	ENZ + ABI + ADT APA + ABI +ADT			6 M	24		
**Comparative studies**												
Sterling [29]	NCT02949284	2020	Phase II RCT	Potency at 1 year	High-risk GS 8–10 or PSA > 20	APA	APA + ABI + ADT	RP only	3 M	10 *	7 *	7 *
Bright [30]	ND	2022	Retrospective	Anti-PSMA staining	High-risk N0M0	ENZ + ADT		RP only	6 M	72	35	37
Ilario [31]	NCT02789878	2022	Phase II	Perioperative complications	High-risk GS 8–10 and/or PSA > 20 and/or cT3 (MRI) and/or cN1	ABI ± APA + ADT		RP only	3 M	124	61	63
Ravi [32]	NCT00924469 NCT01547299 NCT02268175	2022	Retrospective IPTW analysis	Time to BCR MFS	Following each RCT’s eligibility criteria	ARSI + ADT		RP only	6 M	371	112	259

PCa: Prostate cancer, NCT: national clinical trial, I: intervention arm, C: control arm, ASCO-GU: American Society of Clinical Oncology-Genitourinary, PSA: prostate-specific antigen, PSAV: PSA velocity, BS: bone scan, GG: Gleason grade, GS: Gleason score, CT: computed tomography, MRI: magnetic resonance imaging, mpMRI: multiparametric MRI, BCR: biochemical recurrence, ADT: androgen deprivation therapy, ARSI: androgen receptor signaling inhibitor, ABI: abiraterone, APA: apalutamide, CBZ: cabazitaxel, ENZ: enzalutamide, DUT: dutasteride, pCR: pathologic complete response, MRD: minimal residual disease, PSM: positive surgical margin, RCB: residual cancer burden, TEAE: treatment-emergent adverse event, IQR: interquartile range, NCCN: National Comprehensive Cancer Network, ND: no data, M: months, PSMA: prostate-specific membrane antigen, MFS: metastasis-free survival, OS: overall survival, IPTW: inverse probability of treatment weighting. * Described as the number of patients included in each arm.

**Table 2 jpm-13-00641-t002:** Summary of pathological responses to neoadjuvant ARSI-based therapies of included phase II clinical trials.

Author and Year	Proportion of High-Risk pts.	Treatment Regimens	Treatment Duration	Total No. of pts.	pCR, n (%)	MRD < 5 mm, n (%)	Combined Pathologic Response (pCR + MRD), n (%)	≥cT3, n (%)	≥ypT3, n (%)
Lee 2022 [22]	67%	APA	3 M	30	0	ND	ND	10 (33)	12 (48)
Taplin 2014 [13]	74%	ABI + ADT	6 M	30	3 (10)	4 (14)	7 (23)	6 (20)	14 (48)
		ADT followed by ABI + ADT	3 M + 3 M	28	1 (4)	0	1 (3.6)	8 (29)	16 (59)
Montgomery 2017 [14]	79%	ENZ + DUT + ADT	6 M	23	1 (4.3)	3 (13) **	4 (17)	6 (24)	14 (61)
		ENZ		25	0	0 **	0	6 (22)	18 (72)
McKay 2019 [15]	87%	ABI + ENZ + ADT	6 M	50	5 (10)	10 (20)	15 (30)	16 (32)	25 (50)
		ENZ + ADT		25	2 (8)	2 (8)	4 (16)	6 (24)	14 (56)
McKay 2021 [17]	94%	APA + ABI + ADT	6 M	55	7 (13)	5 (9.1)	12 (22)	32 (54)	27 (49)
		ABI + ADT		59	6 (10)	6 (10)	12 (20)	41 (69)	34 (58)
Devos 2022 [18]	98%	APA + ADT	3 M	45	0	17 (38) ***	17 (38)	33 (74)	22 (49)
		ADT		40	0	4 (9) ***	4 (9)	32 (73)	32 (73)
Bastos 2022 [19]	100%	APA + ABI + ADT	3 M	31	1 (3.2)	1 (3.2)	2 (6.4)	49 (79)	19 (61)
		ABI + ADT		31	0	2 (6.4)	2 (6.4)		22 (71)
Fleshner 2022 [20]	100%	CBZ + ABI + ADT	3 M	38	2 (5)	14 (39) ****	16 (44)	ND	22 (58)
		ABI + ADT		32	3 (9)	11 (34) ****	14 (43)		19 (59)
Graham 2022 [21]	100%	APA + ABI + ADT + Indomethacin	3 M	20	1 (5)	6 (30) ***	7 (35)	4 (20)	18 (90)

ARSI: Androgen receptor signaling inhibitor, pts.: patients, ADT: androgen deprivation therapy, ABI: abiraterone, APA: apalutamide, CBZ: cabazitaxel, ENZ: enzalutamide, DUT, dutasteride, pCR: pathologic complete response, MRD: minimal residual disease, M: months, ND: no data. ** Defined as MRD < 3 mm. *** Defined as residual cancer burden < 0.25 cm^3^. **** Defined as <5% of prostate volume involved by a tumor.

**Table 3 jpm-13-00641-t003:** Treatment-emergent adverse events and perioperative complications of eligible studies.

First Author	Year	Treatment Regimen		No. of Patients			TEAEs (Any), n (%)		TEAEs (Grade 3), n (%)		Treatment Discontinuation, n (%)		Perioperative Complications(any), n (%)		Perioperative Complications (CD ≥ 3), n (%)	
		I	C	Total	I	C	I	C	I	C	I	C	I	C	I	C
**Comparative studies**																
Taplin [13]	2014	ABI + ADT (24 w)	ADT (12 w) ABI + ADT (12 w)	58	30	28	12 w: 28 (93) 24 w: 30 (100)	12 w: 28 (100) 24 w: 28 (100)	12 w: 4 (13) 24 w: 7 (23)	12 w: 2 (7) 24 w: 9 (32)	12 w: 3 (10) 24 w: 4 (13)	12 w: 0 24 w: 2 (7)	ND		Any unplanned ER visits 1 (3)	Any unplanned ER visits 3 (11)
Montgomery [14]	2017	ENZ + DUT + ADT	ENZ	48	23	25	25/25 (100)	27/27 (100)	6 (24)	3 (11)	0	0	ND			
McKay [15]	2019	ABI + ENZ + ADT	ENZ + ADT	75	50	25	Hypertension: 16 (32) ALT increase: 17 (34) AST increase: 16 (32)	Hypertension: 6 (24) ALT increase: 1 (4) AST increase: 2 (8)	Hypertension: 5 (10) ALT increase: 5 (10) AST increase: 5 (10)	Hypertension: 1 (4) ALT increase: 0 AST increase: 0	ND		2/47 * (4.3)	0/24 * (0)	ND	
Efstathiou [16]	2019	ABI + ADT	ADT	65	44	21	44 (100)	21 (100)	17 (39)	5 (24)	5 (11)	0	ND			
McKay [17]	2021	APA + ABI + ADT	ABI + ADT	118	59	59	ND		8 (14)	5 (8.5)	ND		Intraoperative 1 (1.8)	Intraoperative 1 (1.8)	Postoperative complications were low and similar between arms	
Devos [18]	2022	APA + ADT	ADT	89	45	40	ND		Grade 3 rash was observed in four (8.9%) patients in the APA + ADT arm		ND		7 (16)	4 (9.1)	1 (2.2)	0
Bastos [19]/ Ilario [31]	2022	APA + ABI + ADT	ABI + ADT	62	31	31	ND 2 grade 5 AEs in the intervention arm		6 (19)	3 (9.7)	ND		30-day complications: 18 (30)		30-day complications: 4 (6.6) Any unplanned ER visits: 7 (12)	
Fleshner [20]	2022	CBZ + ABI + ADT	ABI + ADT	70	38	32	ND		23 (61)	10 (31)	7 (9.1)		ND			
**Single-arm studies**																
Graham [21]	2021	Indomethacin + APA + ABI + ADT		20			Hot flashes: 18 (82) Fatigue: 16 (73) Cognitive changes: 11 (50) Gastrointestinal disorders: 11 (50)		Hypertension: 6 (27) Increased transaminases: 1 (4)		1 (4.8)		No unexpected complications at the time that RP appeared after neoadjuvant therapy			
Lee [22]	2022	APA		30			28 (93)		0		0		5/25 (20)		0	

PCa: Prostate cancer, I: intervention arm, C: control arm, CD: Clavien–Dindo classification, ADT: androgen deprivation therapy, ABI: abiraterone, APA: apalutamide, CBZ: cabazitaxel, ENZ: enzalutamide, DUT: dutasteride, AE: adverse events, TEAE: treatment-emergent adverse event, IQR: interquartile range, ER: emergency room, ND: no data. * Reported as an in-hospital complication.

## Data Availability

No new data was created.

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
