# Peer review of "Neoadjuvant Androgen Receptor Signaling Inhibitors before Radical Prostatectomy for Non-Metastatic Advanced Prostate Cancer: A Systematic Review"

_jpm, 2023, doi:10.3390/jpm13040641_

Round 1

Reviewer 1 Report

the authors explored the interesting and perspective research channel of the Neoadjuvant Androgen Receptor Signaling Inhibitors Before Radical Prostatectomy for Non-metastatic Advanced Prostate Cancer, by conducting a Systematic Review.

The aim of the paper was well stated, the authors explained the research question clearly, the study was well-designed, and the authors presented the results in a clear and repeatable way. The english was fluent and there was no unclear paragraph. The references were adeguate and updated

Reviewer 2 Report

Thank you for inviting me to review the article titled “Neoadjuvant Androgen Receptor Signaling Inhibitors Before Radical Prostatectomy for Non-metastatic Advanced Prostate Cancer: A Systematic Review”

The authors evaluate the current evidence for ARSI before prostatectomy for Non-metastatic advanced prostate cancer.

The data in this review is reported accurately without distortion from the representative papers. I commend the authors for summarizing the current state of evidence for neoadjuvant therapies in non-metastatic prostate cancer.

Comments:

Method: Were there particular years of abstracts reviewed for ASCO or ESMO? Eg. Between 2010-2022 etc.

Line 246 is missing the word ‘gene’ in ERG full form

Line 446: ‘However, although neoadjuvant ADT has never shown a survival benefit compared  to RP alone, most ongoing RCTs set the control arm as neoadjuvant ADT only. To date, the standard of care for high-risk non-metastatic PCa is RP alone when surgical treatment  is applied. The results from this RCT will provide novel insight into the efficacy of  ARSI-based neoadjuvant therapy for non-metastatic advanced PCa, while interpretation may be controversial. ‘ - I agree with this.
